# Investigating the drivers for antibiotic use and misuse amongst medical undergraduates–perspectives from a Sri Lankan medical school

Yasasuru Jayawardhana[1][☉], Avanthi Premaratne[2][‡], Sudeepa Kalpani[1][☉], Sawindya Jayasundara[1][☉], Gihan Jayawardhane[1][☉], Chamini Jayawarna[1][☉], Sarala Gamage[1][☉], Kalana Jayawardhana[1][☉], Radshana Johnsan[1][☉], Chasith Jayasundara[1][☉], Veranja Liyanapathirana[2][‡]*

**1** Faculty of Medicine, University of Peradeniya, Sri Lanka, Sri Lanka, **2** Department of Microbiology, Faculty of Medicine, University of Peradeniya, Sri Lanka, Sri Lanka

☉ These authors contributed equally to this work.
‡ AP and VL also contributed equally to this work.
* veranja.liyanapathirana@med.pdn.ac.lk

**Data Availability Statement:** All relevant data are within the paper and Supporting Information files.

## Abstract

Medical undergraduates are a unique group who gain the theoretical knowledge on prescribing antibiotics but are not authorized to prescribe till full licensure. This unique situation may result in self-medication and unauthorized prescription of antibiotics. This cross-sectional study was conducted among medical students of the Faculty of Medicine, University of Peradeniya, Sri Lanka in 2021 to identify patterns and drivers for antibiotic use and misuse among medical undergraduates. A validated, self-administered Google forms-based online questionnaire was used to gather information on antibiotic use, misuse, and associated factors: demographics, knowledge and perceptions. Two scores; a practice score and a knowledge score were calculated to compare with the associated factors. The study population consisted of 347 medical students with a mean age of 24 (SD1.7) years and 142/347 (40.9%) were male participants. The patterns of misuses identified included; use of antibiotics without a prescription (161/347, 46.4%), keeping left-over antibiotics for future use (111/347, 32.0%), not completing the course of antibiotics (81/347, 23.3%), use of left-over antibiotics (74/347, 21.3%), prescribing to animals (61/347, 17.6%), prescribing antibiotics to family members or friends (51/347, 14.7%), antibiotic self-medication (25/347, 7.2%) and not following the dosage regime prescribed (24/347, 6.9%). The practice score ranged from 33% to 100% (median 87%, IQR 80.0–93.3) and did not differ significantly with either the gender or the year of study. The knowledge score ranged from 4% to 100% (median 87%, IQR: 71.5–95.4) differing significantly according to the year of study. Antibiotic prescription by medical undergraduates was perceived as unacceptable (329/347, 94.8%) by the majority. Individual misuse patterns were associated favourably or unfavourably with gender, year of study, having a health care worker at home and knowledge score. The knowledge score increased with the advancement in training at the medical school while the practice score remained indifferent, highlighting the need to identify the additional drivers of antibiotic misuse among medical undergraduates.

**Funding:** The authors received no specific funding for this work.

**Competing interests:** The authors have declared that no competing interests exist.

# 1. Introduction

Antimicrobial resistance (AMR) is a global health concern leading to significant mortality and morbidity. Murray *et al* in their seminal paper 'Global burden of bacterial antimicrobial resistance in 2019', estimated that 4.95 million deaths were associated with bacterial AMR in 2019 [1]. The burden of AMR is predicted to have disastrous outcomes for low and middle-income countries like Sri Lanka. Antibiotic misuse is the major contributor to emergence of resistant bacteria.

Medical undergraduates are in a unique position in relation to antibiotic use, they gain the theoretical knowledge needed for antibiotic prescription, observe the common prescribing practices in health-care settings, yet are not licensed to prescribe. Furthermore, medical undergraduates are perceived in a positive regard by the public, especially in developing countries. Therefore, their practices of antibiotic use could influence citizens either in a positive or a negative manner.

There is considerable misuse of antibiotics such as over the counter purchase, consumption without prescription, and use of leftover antibiotics, among medical undergraduates worldwide despite the possession of knowledge on antimicrobials and antimicrobial resistance [2, 3]. The main drivers towards these practices show a remarkable difference between countries, indicating that the drivers of antibiotic misuse among this unique category need to be further explored.

We investigated the practices of antibiotic misuse and the main drivers of those practices in a study sample of medical undergraduates of the Faculty of Medicine, University of Peradeniya, Sri Lanka in order to identify target areas for sustainable interventions or for further evaluation.

# 2. Materials and methods

1st, 2nd, 3rd and 4th year medical undergraduates of Sri Lankan citizenry were eligible to participate in this cross sectional study. The batches of 2014/15, 2015/16, 2017/18, 2018/19 academic years of entry were studying in the 4th year, 3rd year, 2nd year and 1st year at the time of data collection, respectively. A self-administered Google form-based questionnaire was designed for the study (S1 Text). This was designed and administered in English. The questionnaire comprised of three main components: section 1- knowledge on antibiotics and antibiotic resistance, section 2- practices on antibiotic use, section 3- attitudes towards antibiotic misuse and antibiotic resistance as a health problem. Closed-ended questions, mainly true/false type and multiple-choice questions were used along with open ended questions.

Review and revision of the questionnaire was performed so that is was easily comprehensible, non-offensive and unbiased, through expert validation by a consultant microbiologist and a specialist in medical education along with pilot testing on a sample of 10 students.

Data were collected during a one month period from 24/05/2021 onwards. The database was monitored daily and accepting answers was disabled once the sample size was fulfilled.

Ethical approval was obtained from Ethics Review Committee of Faculty of Medicine, University of Peradeniya, Sri Lanka (Protocol no 2021/EC/SP/07) and informed written consent was taken through a mechanism inbuilt to the online questionnaire.

The data collection tool itself created the spreadsheet with data. Data were scrutinized to see if all entries are complete. It was secured, and password protected with limited access only for the investigators. Independent variables of the study were gender, age, batch, availability of family members related to health care and dependent variables were practices on antibiotic use, knowledge on antibiotics and antibiotic resistance, and attitudes on antibiotics and antibiotic resistance. The population was described with descriptive statistics. Percentages were

calculated for different patterns of antibiotic use and misuse as well as for different questions on knowledge regarding antibiotics and antibiotic resistance. A practice score and a knowledge score were calculated. (S1 Table). Open ended questions were analyzed thematically.

Distribution of knowledge and practice scores were checked with Shapiro-Wilk test of normalcy. As the data were not normally distributed, the following tests were used for comparisons; categorical variables were compared across groups with Chi-square or Fisher's exact tests, continuous variables were compared across groups with Mann-Whitney U test or Kruskal-Wallis test. A p value of less than 0.05 was considered as significant.

# 3. Results

## 3.1. Overview

During the study, a total of 347 participants were recruited with a mean age of 24.02 (SD 1.7) and a median age of 24.0 (IQR: 23.0–26.9). The response rates for each of the batches were 36.6%, 37.1%, 50.2% and 45.4%. The majority (205/347, 59.1%) were female participants and 108/347 (31.1%) had a family member related to health-care sector at home. Of the 347 participants, 93 (26.8%) were first years, while 103 (29.7%), 76 (21.9%) and 75 (21.6%) were 2nd, 3rd and 4th year students respectively.

## 3.2. Patterns of antibiotic misuse and disposal

The commonest antibiotic misuse identified was using antibiotics without a prescription (161/347, 46.4%). In addition, the following patterns of misuse were identified; keeping left-over antibiotics for future use (111/347, 32.0%), not completing the prescribed course of antibiotics (81/347, 23.3%), use of left-over antibiotics (74/347, 21.3%), prescribing to animals (61/347, 17.6%), prescribing antibiotics to family members or friends (51/347, 14.7%), antibiotic self-medication (25/347, 7.2%) and not following the dosage regime prescribed (24/347, 6.9%).

The ways of getting antibiotics without a prescription was assessed through a multiple-choice question where participants could pick more than one option. The acknowledged methods of obtaining antibiotics without a prescription were; over the counter purchase (127/161, 78.9%), use of left-overs (28/161, 17.4%) and obtaining from friends or family (51/161, 31.7%).

The main reasons given for not consulting a prescriber was given as, having learnt the drugs that would be prescribed (52), assuming to know the drug the doctor would prescribe (39), availability of left-over antibiotics (14), having a family member who is a doctor (3), not having access to a doctor due to the pandemic (2).

On inquiry on practices of taking antibiotics for specified common infections, 39/347 (11.2%) stated that they took antibiotics without prescriptions for sore throat (Table 1).

Regarding the disposal methods for left-over antibiotics, 192/347 (55.3%) stated that they dispose antibiotics with household trash, 12/347 (3.5%) stated that they burn/incinerate, 10/347 (2.9%) mentioned of discarding to the environment, three (0.9%) stated that they had buried leftover antibiotics, two participants (0.6%) had flushed medicines in the toilet, one (0.3%) participant had handed over the left-over antibiotics to the hospital to be discarded whereas another one (0.3%) participant had put them into flower pots.

**3.2.1. Last recalled instance of antibiotic use.** Of the participants, 341 (98.3%) gave information on the indication for the last recalled instance of taking antibiotics. The commonest reason given was sore throat (61, 17.6%), followed by common cold (47, 13.5%) and wound infection (24, 6.9%). Further, regarding the last recalled use of antibiotics, 299 (86.2%) had taken antibiotics with prescription whereas 40 (11.5%) had taken without a prescription. Of those who had taken antibiotics with a prescription, 241/299 (80.6%) had been prescribed

**Table 1. Antibiotic use for common conditions.**

| Condition | Have not taken antibiotics (n, %) | Taken with prescription (n, %) | Taken without prescription (n, %) |
|---|---|---|---|
| Common cold | 193 (55.6%) | 121 (34.9%) | 33 (9.5%) |
| Sore throat | 133 (38.3%) | 175 (50.4%) | 39 (11.2%) |
| Non-specific fever | 254 (73.2%) | 83 (23.9%) | 10 (2.9%) |
| Headache | 325 (93.7%) | 19 (5.5%) | 3 (0.9%) |
| Wound infections | 162 (46.7%) | 158 (45.5%) | 27 (7.8%) |
| Urinary tract infection | 250 (72.0%) | 94 (27.1%) | 3 (0.9%) |
| Diarrhoea | 225 (64.8%) | 108 (31.1%) | 14 (4.0%) |

antibiotic after a medical examination by a qualified doctor, 15/299 (5.0%) had been prescribed after a telephone conversation with a qualified doctor, 19/299 (6.4%) had been prescribed antibiotics by a family member who is a medical professional while only 22/299 (7.4%) stated that diagnostic tests were performed before the antibiotics were prescribed.

### 3.3 Practice score

The practice score distribution showed a mean of 86.4 (SD 12.1) and a median of 86.7 (IQR 80.0–93.3). Practice score did not differ significantly according to the batches, gender, or having a family member related to healthcare sector at home (Table 2).

### 3.4 Knowledge on antibiotics and antibiotic resistance

Majority (290/347, 83.6%) of the participants defined antibiotics correctly as drugs that act against specific bacteria. However, 53/347 (15.3%) identified antibiotics as drugs that can act against all micro-organisms, while three of the 347 (0.9%) identified antibiotics as drugs that can be used in any illness. Of the participants, 328/347 (94.5%) had stated they knew what "antibiotic resistance" meant, while 19/347 (5.5%) stated that they did not.

Responses to the open-ended question on defining antibiotic resistance were categorized into themes. The commonest theme identified was "Bacteria lose sensitivity, develop resistance against (certain) antibiotics"; ideas falling within this theme was given by 118/347 (34.0%) participants. The other common themes and the number of participants giving similar views are as follows; "previously susceptible bacteria; cannot be killed or controlled due to evolution, development of mechanisms, acquiring mutations to develop resistance"–expressed by 103/347 (29.7%) participants and "using antibiotics for a long time, wrong dose, misuse, irrational

**Table 2. Distribution of practice score according to the batches, gender and having a family member related to the healthcare sector.**

| Category | | Mean practice score (SD) | Median practice score (IQR) | Significance (2-sided) |
|---|---|---|---|---|
| Year of study | 4th year (n = 75) | 88.5 (9.4) | 93.3 (80.0–93.3) | 0.095^ |
| | 3rd year (n = 76) | 87.3 (12.7) | 93.3 (80.0–93.3) | |
| | 1st year (n = 103) | 86.9 (11.7) | 86.7 (80.0–100.0) | |
| | 1st year (n = 93) | 83.5 (13.5) | 86.7 (73.3–93.3) | |
| Gender | Female | 86.9 (11.5) | 93.3 (80.0–93.3) | 0.545* |
| | Male | 85.7 (12.9) | 86.7 (73.3–95.0) | |
| Having a family member related to the healthcare sector | No | 86.5 (12.5) | 93.3 (80.0–93.3) | 0.542* |
| | Yes | 86.2 (11.2) | 86.7 (80.0–93.3) | |

^ Independent-Samples Kruskal-Wallis Test

* Independent-Samples Mann-Whitney U test

**Table 3. Knowledge on antibiotics and AMR.**

| Stem | | Agree/strongly agree (n %) | Disagree/strongly disagree (n %) | Do not know, not answered (n %) |
|---|---|---|---|---|
| I. Antibiotics are medicine that fight infections caused by bacteria in humans and; animals by either killing the bacteria or inhibiting their multiplication | | 344 (99.1) | 1 (0.3%) | 1 (0.6%) |
| II. Antibiotic resistance happens when bacterial strains develop ability to withstand the effect of drugs used to treat infections caused by them | | 329 (94.8%) | 3 (0.9%) | 15 (14.3%) |
| III. Antibiotic resistance means that the human body is becoming resistant to antibiotics | | 70 (20.2%) | 239(8.8%) | 38 (11.0%) |
| IV. Antibiotics are essential for the following conditions: | (a) Common cold & flu | 77 (22.2%) | 248 (71.5%) | 22 (6.3%) |
| | (b) Any sore throat | 113 (32.6%) | 195 (56.2%) | 39 (11.2%) |
| | (c) All fevers | 20 (5.8%) | 284 (81.8%) | 43 (12.4%) |
| | (d) To relieve bodily pain | 22 (6.3%) | 292 (84.1%) | 33 (9.5%) |
| | (e) Headache | 27 (7.8%) | 289 (83.3%) | 31 (8.9%) |
| | (f) Urinary tract infections (not considered for marking) | 325 (93.7%) | 9 (2.6%) | 13 (3.7%) |
| | (g) Skin and soft tissue infections (not considered for marking) | 307 (88.5%) | 12 (3.5%) | 28 (8.1%) |
| | (h) Strep throat | 278 (80.1%) | 8 (2.3%) | 61 (17.6%) |
| | (i) All cases of vomiting/diarrhoea | 25 (7.2%) | 273 (78.7%) | 49 (14.1%) |
| | (j) As prophylaxis for some specific infection | 236 (68.0%) | 18 (5.2%) | 93 (26.8%) |
| V. The following actions contribute to emergence of antibiotic resistance | (a) Using antibiotics without prescription | 315 (90.8%) | 22 (6.3%) | 10 (2.9%) |
| | (b) Taking antibiotics for self-limiting diseases | 281 (81.0%) | 12 (3.5%) | 54 (15.6%) |
| | (c) Over-prescription of antibiotics by healthcare professionals in some clinics | 303 (87.3%) | 11 (3.2%) | 33 (9.5%) |
| | (d) Lack of hygiene and poor sanitation | 129 (37.2%) | 90 (25.9%) | 128 (36.9%) |
| | (e) Poor infection control in hospitals and clinics | 243 (70.0%) | 35 (10.1%) | 69 (19.9%) |
| | (f) Incomplete treatments with antibiotics | 303 (87.3%) | 12 (3.5%) | 32 (9.2%) |
| | (g) Overuse of antibiotics in animals | 248 (71.5%) | 14 (4.0%) | 85 (24.5%) |
| VI. Antibiotic resistance affect the people as follows: | (a) Prolonged morbidity | 286 (82.4%) | 4 (1.2%) | 57 (16.4%) |
| | (b) Prolonged hospitalization | 315 (90.8%) | 3 (0.9%) | 29 (8.4%) |
| | (c) Risk of mortality | 307 (88.5%) | 3 (0.9%) | 37 (10.7%) |
| | (d) Increased medical costs | 316 (91.1%) | 5 (1.4%) | 26 (7.5%) |

use lead to resistance"–expressed by 51/347 (14.7%) participants. Interestingly, 9/347 (2.6%) of the participants responded to say that the "human body develops resistance to antibiotics". On further analysis of these answers, 285/347 (82.1%) participants were found to have given correct answers while 62/347 (17.9%) had stated incorrect answers.

Close ended questions asked and responses related to the knowledge on antibiotics and antimicrobial resistance is given in Table 3. Interestingly, 70/347 (20.2%) participants agreed that antibiotic resistance results when humans become resistant to antibiotics, 77/347 (22.2%) agreed that antibiotics are essential for common cold and flu while only 129/347 (37.2%) recognized that poor sanitation and hygiene contributed to emergence of resistance.

### 3.5. Knowledge score

The total marks allocated for the knowledge on antibiotics and antibiotic resistance were 100 and it ranges from a minimum score of 4 to a maximum score of 100 with a mean of 81.4 (SD 19.0) and a median of 87.0 (IQR– 71.5–95.4). Knowledge score of the participants were analyzed according to the batches, gender and presence of a family member related to healthcare sector at home. Results showed that there is a significant difference in median knowledge score across

**Table 4. Distribution of knowledge score according to the year of study, gender and having a family member related to the healthcare sector.**

| Category | | Mean practice score (SD) | Median practice score (IQR) | Significance (2-sided) |
|---|---|---|---|---|
| **Year of study** | 4th year | 93.0 (6.6) | 95.7 (91.3–100.0) | <0.001^ |
| | 3rd year | 93.4 (7.7) | 95.7 (91.3–100.0) | |
| | 2nd year | 83.6 (13.8) | 87.0 (78.3–95.7) | |
| | 1st year | 59.9 (18.9) | 60.9 (47.8–73.9) | |
| **Gender** | Female | 81.3 (18.6) | 87.0 (69.6–95.7) | 0.642* |
| | Male | 81.7 (19.6) | 91.3 (69.6–95.7) | |
| **Having a family member related to the healthcare sector** | No | 80.6 (19.1) | 87.0 (69.6–95.7) | 0.101* |
| | Yes | 83.3 (18.6) | 91.3 (78.3–95.7) | |

^according to the Independent-Samples Kruskal-Wallis Test

*according to the Independent-Samples Mann-Whitney U test

the batches (Table 4). There was no significant difference in median knowledge score according to gender or the presence of a family member related to healthcare sector at home.

## 3.6. Perceptions on antibiotic misuse

Eighteen (5.2%) students stated that they thought antibiotic prescription by medical students is acceptable while 329 (94.8%) thought it is not. Thematic analysis was done to identify the possible reasoning for these attitudes and common themes are given in Table 5.

Seventeen participants (4.9%) had agreed that it was acceptable to take antibiotics without seeing a qualified doctor whereas 316 (91.1%) had disagreed. Of the respondents, 296 (85.3%) had agreed that antibiotic over-prescription contributes to the development of antibiotic resistance. Also, 193 participants (55.6%) had agreed that their knowledge on antibiotics and antibiotic resistance was not adequate. Other responses related to confidence and perception on antibiotic use is given in Table 6.

**Table 5. Themes for reasons in justifying or not justifying antibiotic prescription by medical students.**

| | Justification and reasons | Frequency n, % |
|---|---|---|
| Antibiotic prescription by medical students is **acceptable** | (a) Acceptable to prescribe for some infections as we have learned about antibiotics | 11 (3.2%) |
| | (b) Acceptable with a diagnosis, following guidelines and formularies etc. | 3 (0.9%) |
| | (c) Can prescribe depending on the site of the infection | 1 (0.3%) |
| | (d) Doctor also prescribed the same drug previously | 1 (0.3%) |
| | Total | 16 (4.6%) |
| Antibiotic prescription by medical students is **not acceptable** | (a) Not yet qualified, unethical, illegal, need MBBS, not licensed | 115 (3.1%) |
| | (b) Have to confirm bacterial infection and ABST to prevent resistance and need a proper medical examination | 10 (2.9%) |
| | (c) It can contribute to antibiotic resistance | 21 (6.1%) |
| | (d) Can misdiagnose due to lack of knowledge, expertise | 89 (25.6%) |
| | (e) Lack of clinical exposure | 9 (2.6%) |
| | (f) Other | 10 (2.9%) |
| | Total | 254 (73.2%) |
| Not answered | - | 77 (22.2%) |
| Total | | 347 (100.0%) |

**Table 6. Responses related to confidence and perception on antibiotics use.**

| | Agree/ Strongly agree n, % | Disagree/ Strongly disagree n,% | Do not know/ Not answered n, % |
|---|---|---|---|
| Do you think it is acceptable to take antibiotics without seeing a qualified doctor? | 17 (4.9%) | 316 (91.1%) | 14 (4.0%) |
| It is safe to use antibiotics whenever we think they are needed | 22 (6.3%) | 308 (88.8%) | 17 (4.9%) |
| Do you agree that antibiotic over prescription contributes to the development of resistance? | 296 (85.3%) | 16 (4.6%) | 35 (10.1%) |
| It is essential to give important facts to the patients while prescribing antibiotics in order to prevent antibiotics misuse and over use | 333 (96.0%) | 3 (0.9%) | 11 (3.2%) |
| It is good to prescribe broad spectrum antibiotics at the beginning of antibiotic treatment indiscreetly | 112 (32.3%) | 235 (67.7%) | 0 (0.0%) |
| I think I have sufficient knowledge on antibiotic use | 120 (34.6%) | 161 (46.4%) | 66 (19.0%) |
| My knowledge on antibiotics and antibiotic resistance is adequate | 154 (44.4%) | 193 (55.6%) | 0 (0.0%) |
| I know when antibiotics are needed | 220 (63.4%) | 127 (36.6%) | 0 (0.0%) |
| I have taken greater precautions when using antibiotics after learning about them | 309 (89.0%) | 38 (11.0%) | 0 (0.0%) |
| I've informed family and friends about antibiotic resistance | 259 (74.6%) | 88 (25.4%) | 0 (0.0%) |

### 3.7. Factors associated with antibiotic misuse

Some practices were more prevalent among males while some were more prevalent among females. A significantly higher proportion of male participants (76/142, 53.5%) than female participants (85/205, 41.5%) had used antibiotics without a prescription (p = 0.029, Pearson Chi-square test). Similarly, a significantly higher proportion of male participants (38/142, 26.8%) than female participants (36/205, 17.6%) had used left over antibiotics on themselves and on others (p = 0.046, Pearson Chi-Square test). However, more female participants (61/205, 29.8%) than male participants (20/142, 14.1%) mentioned that they generally do not complete a full course of antibiotics. (p = 0.001, Pearson Chi-Square test). Interestingly, participants who had a family member working in health-care work were less likely to prescribe antibiotics to others as opposed to who did not (8/108, 7.4% vs 43/239, 18.0%; p = 0.013, Pearson Chi-Square test). However, those with a family member working in healthcare were more likely to prescribe antibiotics to animals as opposed to those who did not (28/108, 25.9% vs 33/239, 13.8%; p = 0.007, Pearson Chi-Square test).

When analyzed according to the year of study, the 4th year students were more likely to complete the full course of antibiotics, and take the drugs according to the prescribed regime, however, they had also prescribed antibiotics more than the other batches. Further, those who generally complete the full course of antibiotics and take according to the prescribed dosage regime had a higher knowledge score. At the same-time, those who prescribed antibiotics animals, also had a higher knowledge score (S2 Table).

## 4. Discussion

This cross-sectional study was aimed at identifying the current practices leading to antibiotic misuse and drivers for antibiotic misuse among medical undergraduates of a Sri Lankan University. We had response rates that are comparable and alike to those reported elsewhere on medical students and doctor [4].

The main uses of antibiotics in this sample of medical students were for sore throat, wound infections, common cold and diarrhea. Globally and locally in Sri Lanka, respiratory tract infections are major contributors for antibiotic misuse even among medical students [5, 6].

The common antibiotic misuse patterns identified in the study were consumption of antibiotics without prescription, preserving left-over antibiotics for future use and not completing a

full course of antibiotics. Globally, similar patterns of misuse have been identified [2, 3, 7, 8]. Self-medication with antibiotics has also been found as a prevalent practice among medical students [2, 3, 8]. Antibiotic self-medication is facilitated by availability of previous prescriptions and confidence in one's theoretical knowledge on antibiotic use. In our study too, a considerable percentage (46.4%) acknowledged on taking antibiotics without prescription.

The highest percentage of antibiotic use without prescription was for treatment of sore throat, followed by common cold. Antibiotic prescription for respiratory symptoms is a common practice even by qualified doctors [9]. This is partly driven by the fear of "bacterial super-infections" and a perceived experience that antibiotics do speed up recovery [10].

It was concerning to note that antibiotic use as a whole, and antibiotic use without a prescription were both most commonly for respiratory tract infections, that are most likely to be of viral origin. Further, given that this trend continues towards clients after qualifying [9], it is imperative that action is taken to correct this. This process should involve novel methods to include highlighting the importance of unlearning or letting go of some strongly held beliefs.

We assessed the methods of disposal of left-over antibiotics. Most participants had disposed the antibiotics with household trash. The practice is possibly due to the convenience of disposal and establishes a case for implementing a streamlined collection method for any left-over medicines before disposal as antibiotic residues are found in the environment plays a substantial role in increasing the burden of antibiotic resistance [11].

Overall, the practice score had a median of 86.7 (IQR 80.0–93.3) indicative of a positive deviation towards standard practices. However, there was no significant difference in practice scores between the participants from different study years. We expected the participants undergoing clinical training with more theoretical knowledge as well as clinical exposure on antibiotics to have better practice scores. Antibiotic prescription in hospitals in Sri Lanka leaves much to be desired [12]. This is particularly so for respiratory tract infections [13]. Thus, students who gain the theoretical knowledge on correct antibiotic use do not see the theory in practice. Instead, they see incorrect practices in their clinical training, thus, despite having the appropriate knowledge, students are likely to follow incorrect practices. This could explain why an improvement of practice score was not seen as the students progressed in their studies.

The practice score did not differ according to the gender or having a healthcare sector-related family member at home. These findings are somewhat different to findings from other countries [14, 15]. We hypothesize that antibiotic use, has a socio-cultural element to it. Therefore, factors associated with different patterns of misuse are influenced by socio-cultural determinants at a given locale and would vary from country to country and even within communities in countries.

Of the respondents, about 15% agreed that antibiotics were drugs that act against all microorganisms. A study done in UAE showed that about a third of the medical students were found to have similar views [16]. This misconception may be stemming from the misuse of antibiotics for common viral infections, that medical students see as children, and as mentioned earlier, it needs to be "unlearnt" at the medical school.

A further misconception, despite being found less frequently in our study (2.6%) was the belief that antibiotic resistance means that the human body becomes resistant to antibiotics. This misconception has been found frequently among general public [17]. This again, should be corrected through education.

The knowledge score differed significantly across batches with both third- and fourth-year students having the highest median knowledge scores compared to first- and second-year students. This has been shown in many countries and is the expected norm as both theoretical knowledge on antibiotics and infections as well as the clinical experience to consolidate the

knowledge should improve as the students progress through years of training [18, 19]. However, we recommend the key conceptual definitions in infections, antibiotics and antimicrobial resistance to be introduced as early as possible with re-enforcement in subsequent years.

Perceptions and attitudes on antibiotic misuse is focal for the prevention of antibiotic misuse. Most of the participants agreed that antibiotic over prescription contributes to development of antibiotic resistance, which is a favourable attitude. However, we did not assess what exactly is interpreted as misuse among the study participants. Less than one third of the participants agreed that they have sufficient knowledge on antibiotics. It has been recently found that graduates expressed les confidence in prescribing antibiotics than medications for other ailments [20]. Similarly in a study done in an urban medical school in the northeast United States had also shown that more than 75% of the students preferred more education on antibiotics [21]. The expanding content of medical curricula has reduced the time available per subject, therefore alternative methods to disseminate knowledge on antibiotic use, such as fixed learning modules need to be explored.

Of the participants 18 (5.2%) had claimed that antibiotic prescription by medical students is acceptable. Even though the percentages are seemingly small, it is important to intervene for a positive change in perceptions. In addition to giving the proper knowledge on antibiotic use, this may require a more humanistic approach, where medical undergraduates are made to reflect up on their wider responsibility towards the society in terms of antibiotic resistance.

We identified that the practice scores did not differ among the categories of participants that we compared. However, when it came to individual practices, differences were noted among the categories studied. Of interest was to note that male students tend to self-medicate more frequently than females and use leftover antibiotics while females were less likely to complete a full course of antibiotics. Previous studies have identified that female students tend to self-medicate more frequently and authors have attributed this to better attitudes towards self-care among females to this practice [22]. However, we hypothesize that the use of antibiotics as a "quick fix" may be the reason why more male students are driven to self-medicate [23].

Further, we also found that certain malpractices became commoner as the medical students progressed in their undergraduate career, and some practices improved. This could be attributed to the gaining of knowledge and confidence as well as seeing more of the real-world practice. We found that those with a family member related to health-care were less likely to self-medicate or to prescribe antibiotics for others. It may be these students tend to have easier access to a valid prescription than other students, and therefore are likely to prescribe less. However, we did not assess if the family member is a prescriber or belonged to another category of health-care workers.

We also identified that those who complete a full course of antibiotics and take antibiotics according to the prescribed dosage regime had higher knowledge. This highlights the important role knowledge itself plays on practices. Even though other factors also influence antibiotic use, the importance of adequate knowledge cannot be belittled.

In conclusion, we identified that despite the satisfactory level of knowledge and the seemingly positive perceptions, practices related to antibiotic use failed to show an improvement with the progress of study years, and that different antibiotic misuse patterns are associated with different factors such as gender, knowledge as well as family circumstances. We recommend that standardized methods of evaluation be developed to assess a medical student's knowledge and perception on antibiotics and antimicrobial resistance be included to the medical curricula. So that in addition to just providing knowledge, features that drive practice such as attitudes are also addressed. Reflective activities that emphasize the wider role a medical student has in antibiotic use and misuse may also be helpful in changing the practices. Unless this

is done, this generation of medical students are also likely to continue to engage in practices such as prescribing antibiotics for common viral infections.

### 4.1. Limitations

Our study was conducted among medical students of a single university. However, all medical students in Sri Lanka are selected from a competitive Advanced/Level examination, all faculties prepare students for a common examination at exit, and work towards a common subject benchmark statement. Therefore, the population is relatively homogenous and are taught similar curricula, enabling generalization to all medical schools. We did not calculate a sample size and used convenience sampling for the study, given the circumstances of COVID-19 pandemic. However, we have a relatively larger sample size of 347.

## Supporting information

**S1 Text. Questionnaire.**
(DOCX)

**S1 Table. Marking scheme for the questionnaire.**
(DOCX)

**S2 Table. Factors associated with antibiotic misuse.**
(DOCX)

**S1 Data. Database.**
(XLSX)

## Acknowledgments

The Faculty of Medicine, University of Peradeniya is acknowledged for permitting to conduct the study.

## Author Contributions

**Conceptualization:** Yasasuru Jayawardhana, Sudeepa Kalpani, Sawindya Jayasundara, Gihan Jayawardhane, Chamini Jayawarna, Sarala Gamage, Radshana Johnsan, Chasith Jayasundara, Veranja Liyanapathirana.

**Data curation:** Yasasuru Jayawardhana, Avanthi Premaratne, Sudeepa Kalpani, Sawindya Jayasundara, Gihan Jayawardhane, Chamini Jayawarna, Sarala Gamage, Kalana Jayawardhana, Radshana Johnsan, Chasith Jayasundara.

**Formal analysis:** Yasasuru Jayawardhana, Avanthi Premaratne, Gihan Jayawardhane, Veranja Liyanapathirana.

**Investigation:** Yasasuru Jayawardhana, Sudeepa Kalpani, Sawindya Jayasundara, Gihan Jayawardhane, Chamini Jayawarna, Sarala Gamage, Kalana Jayawardhana, Radshana Johnsan, Chasith Jayasundara.

**Methodology:** Yasasuru Jayawardhana, Sudeepa Kalpani, Sawindya Jayasundara, Gihan Jayawardhane, Chamini Jayawarna, Sarala Gamage, Kalana Jayawardhana, Radshana Johnsan, Chasith Jayasundara, Veranja Liyanapathirana.

**Project administration:** Yasasuru Jayawardhana, Veranja Liyanapathirana.

**Supervision:** Veranja Liyanapathirana.

**Validation:** Avanthi Premaratne, Veranja Liyanapathirana.

**Writing – original draft:** Avanthi Premaratne.

**Writing – review & editing:** Yasasuru Jayawardhana, Avanthi Premaratne, Sudeepa Kalpani, Sawindya Jayasundara, Gihan Jayawardhane, Chamini Jayawarna, Sarala Gamage, Kalana Jayawardhana, Radshana Johnsan, Chasith Jayasundara, Veranja Liyanapathirana.

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
