## [Decision Letter · Decision Letter 0]

12 Dec 2022

PGPH-D-22-01704

Antibiotic use, misuse, and drivers of antibiotic misuse among medical undergraduates - a Sri Lankan experience

Dear Dr. Liyanapathirana,

Thank you for submitting your manuscript to PLOS Global Public Health. After careful consideration, we feel that it has merit but does not fully meet PLOS Global Public Health’s publication criteria as it currently stands. Therefore, we invite you to submit a revised version of the manuscript that addresses the points raised during the review process.

We look forward to receiving your revised manuscript.

Kind regards,

Dione Benjumea-Bedoya, Ph.D

Guest Editor

Journal Requirements:

1. Please change "female” or "male" to "woman” or "man" as appropriate, when used as a noun (see for instance https://apastyle.apa.org/style-grammar-guidelines/bias-free-language/gender).

2.  PLOS Global Public Health does not copy edit accepted manuscripts (https://journals.plos.org/globalpublichealth/s/criteria-for-publication#loc-5). To that effect, please ensure that your submission is free of typos and grammatical errors.

3. Please send a completed 'Competing Interests' statement, including any COIs declared by your co-authors. If you have no competing interests to declare, please state "The authors have declared that no competing interests exist". Otherwise please declare all competing interests beginning with the statement "I have read the journal's policy and the authors of this manuscript have the following competing interests:"

4. We have noticed that you have a list of Supporting Information legends "S1 Text. Google forms based self administered questionnaire" in your manuscript. However, there are no corresponding files uploaded to the submission. Please upload them as separate files with the item type 'Supporting Information'.

5. In the online submission form, you indicated that "The database would be made available to any researcher who has a legitimate requirement, through the corresponding author". All PLOS journals now require all data underlying the findings described in their manuscript to be freely available to other researchers, either 1. In a public repository, 2. Within the manuscript itself, or 3. Uploaded as supplementary information.

Additional Editor Comments (if provided):

This is an interesting work related to the use of antibiotics among medical undergraduates, however, the manuscript needs to be copyedited and it is necessary to improve the structure of the manuscript specially results and discussion.

Reviewers' comments:

Reviewer's Responses to Questions

**Comments to the Author**

1. Does this manuscript meet PLOS Global Public Health’s publication criteria? Is the manuscript technically sound, and do the data support the conclusions? The manuscript must describe methodologically and ethically rigorous research with conclusions that are appropriately drawn based on the data presented.

Reviewer #1: Yes

Reviewer #2: Yes

2. Has the statistical analysis been performed appropriately and rigorously?

Reviewer #1: Yes

Reviewer #2: N/A

3. Have the authors made all data underlying the findings in their manuscript fully available (please refer to the Data Availability Statement at the start of the manuscript PDF file)?

Reviewer #1: No

Reviewer #2: Yes

4. Is the manuscript presented in an intelligible fashion and written in standard English?

Reviewer #1: Yes

Reviewer #2: Yes

5. Review Comments to the Author

Reviewer #1: This is an interesting and novel study that looks at how medical students, who are at various stages of their training in infectious diseases, antibiotic biology and use, and clinical training, use or misuse antibiotics. The study addresses their knowledge, which as expected does change as they progress in medical education, and their practice, which does not. This “knowledge-practice” gaps is particularly interesting.

The study is well done but a few factors should be addressed.

• The questionnaire needs to be provided, either in supplemental data or a fully accessible site. That allows others to use the information to see how this “knowledge-practice” gap varies globally.

• The data availability does not appear to meet PLoS guidelines.

• The Discussion largely just repeats the Results and does not integrate broader social science studies that have examined this gap between what individuals know and what they do. There is a rich scholarly literature on this and the authors are strongly encouraged to incorporate it into the Discussion in lieu of just repeating the Results. Building on the basis of why individuals do this will be fundamental in addressing this “know-do” gap.

Reviewer #2: Thank you for this manuscript.

My comments are in the order they appear in the manuscript.

The title: the word misuse is repetitive, consider changing title to 'Investigating the drivers for antibiotic use and misuse amongst medical undergraduates - perspectives from Sri Lankan medical undergraduates

Abstract

Needs to be rewritten to clarify a few things. Are you suggesting the medical student cohorts will misuse antibiotics for themselves? Since they cannot yet prescribe.

Remove the word 'done' the study was conducted across xyz

And again repetition of the word misuse please rewrite as per title.

How the results are presented are not coherent. Add ratios as well as %s. 347 students with a mean age of 24, of whom 142/347 (40.9%) were male.

The rest of results need to be rewritten for clarity. It is not clear how knowledge was scored.

Introduction

AMR in brackets in first line. An assumption is being made that medical undergraduates gain theoretical knowledge on AMR. Is there any data from Sri Lanka on how much is taught on AMR and prescribing as part of undergraduate curricula?

Page 3 line 57 are you referring to Sri Lanka specifically?

Methods

What were the steps to ensure the questionnaire content was comprehensible, non-offensive, unbiased?

Please provide the questionnaire as a supplementary file. Data are plural hence data were not data was

Was a calculation made for an appropriate sample size and response rate? If so how and if not why not?

How was knowledge assessed?

What does throwing to the environment mean?

How was significance assessed? E.g. you need to describe in methods. And are your data powered to apply statistical analyses? What hypothesis are you proving or disproving?

It is not clear to me to what aspect of the data the thematic analysis refers to. This needs a section in methods and justification of thematic analysis to quantitative data. Or is this to the free text data?

This study is not powered for such representation of data. Thirteen tables for a study including a survey of 347 participants is not really appropriate. I suggest you review how the data are presented and select as a maximum four tables with key results that are meaningful and relate to your objectives. Same goes for all the power calculations.

Discussion

What is a satisfactory response rate? Also no need to repeat results again in discussion. Do not repeat the results in discussion - rather try and interpret what they mean and what implications they have for addressing AMR in context of Sri Lanka. Discussion needs to be rewritten to remove all results and include a meaningful interpretation of the implications. As well as being shortened significantly and remove the sub headings.

This manuscript also needs a limitations section.

6. PLOS authors have the option to publish the peer review history of their article (what does this mean?). If published, this will include your full peer review and any attached files.

**Do you want your identity to be public for this peer review?** For information about this choice, including consent withdrawal, please see our Privacy Policy.

Reviewer #1: No

Reviewer #2: No

---

## [Decision Letter · Decision Letter 1]

14 Feb 2023

PGPH-D-22-01704R1

Investigating the drivers for antibiotic use and misuse amongst medical undergraduates – perspectives from a Sri Lankan medical school

Dear Dr. Liyanapathirana,

Thank you for submitting your manuscript to PLOS Global Public Health. After careful consideration, we feel that it has merit but does not fully meet PLOS Global Public Health’s publication criteria as it currently stands. Therefore, we invite you to submit a revised version of the manuscript that addresses the points raised during the review process.

EDITOR:

The reviewers' comments have been addressed but the manuscript needs to be copyedited.

We look forward to receiving your revised manuscript.

Kind regards,

Dione Benjumea-Bedoya, Ph.D

Guest Editor

Journal Requirements:

Additional Editor Comments (if provided):

Thank you very much for the revised version of the manuscript, most of the reviewers' comments were addressed. However, to consider this manuscript suitable for publication it is necessary to copyedit it.

Reviewers' comments:

Reviewer's Responses to Questions

**Comments to the Author**

1. If the authors have adequately addressed your comments raised in a previous round of review and you feel that this manuscript is now acceptable for publication, you may indicate that here to bypass the “Comments to the Author” section, enter your conflict of interest statement in the “Confidential to Editor” section, and submit your "Accept" recommendation.

Reviewer #1: All comments have been addressed

Reviewer #2: All comments have been addressed

2. Does this manuscript meet PLOS Global Public Health’s publication criteria? Is the manuscript technically sound, and do the data support the conclusions? The manuscript must describe methodologically and ethically rigorous research with conclusions that are appropriately drawn based on the data presented.

Reviewer #1: Yes

Reviewer #2: Yes

3. Has the statistical analysis been performed appropriately and rigorously?

Reviewer #1: Yes

Reviewer #2: N/A

4. Have the authors made all data underlying the findings in their manuscript fully available (please refer to the Data Availability Statement at the start of the manuscript PDF file)?

Reviewer #1: Yes

Reviewer #2: Yes

5. Is the manuscript presented in an intelligible fashion and written in standard English?

Reviewer #1: Yes

Reviewer #2: Yes

6. Review Comments to the Author

Reviewer #1: The authors have addressed the comments satisfactorily.

Reviewer #2: Thank you for addressing the comments.

There are still some typos that the production team will pick up and will require correction. For example line 84 - should read 'that it was...'

7. PLOS authors have the option to publish the peer review history of their article (what does this mean?). If published, this will include your full peer review and any attached files.

**Do you want your identity to be public for this peer review?** For information about this choice, including consent withdrawal, please see our Privacy Policy.

Reviewer #1: **Yes: **Guy Hughes Palmer

Reviewer #2: **Yes: **ESMITA CHARANI

---

## [Editor Report · Decision Letter 2]

27 Feb 2023

Investigating the drivers for antibiotic use and misuse amongst medical undergraduates – perspectives from a Sri Lankan medical school

PGPH-D-22-01704R2

Dear Dr/ Liyanapathirana,

We are pleased to inform you that your manuscript 'Investigating the drivers for antibiotic use and misuse amongst medical undergraduates – perspectives from a Sri Lankan medical school' has been provisionally accepted for publication in PLOS Global Public Health.

Best regards,

Dione Benjumea-Bedoya, Ph.D

Guest Editor

Thank you for the revised version of the manuscript and for taking into account the recommendation of copyediting.